# The Impact of the COVID-19 Outbreak on Mental Wellbeing in Children with a Chronic Condition Compared to Healthy Peers

**DOI:** 10.3390/ijerph19052953

**Published:** 2022-03-03

**Authors:** Johanna W. Hoefnagels, Annelieke B. Schoen, Sabine E. I. van der Laan, Lyan H. Rodijk, Cornelis K. van der Ent, Elise M. van de Putte, Geertje W. Dalmeijer, Sanne L. Nijhof

**Affiliations:** 1Department of Paediatrics, Wilhelmina Children’s Hospital, University Medical Center Utrecht, Utrecht University, 3584 EA Utrecht, The Netherlands; a.b.schoen-2@umcutrecht.nl (A.B.S.); S.E.I.vanderLaan-4@umcutrecht.nl (S.E.I.v.d.L.); L.H.Rodijk@umcutrecht.nl (L.H.R.); e.vandeputte@umcutrecht.nl (E.M.v.d.P.); s.l.nijhof@umcutrecht.nl (S.L.N.); 2Department of Pediatric Pulmonology, Wilhelmina Children’s Hospital, University Medical Center Utrecht, Utrecht University, 3584 EA Utrecht, The Netherlands; k.vanderent@umcutrecht.nl; 3Julius Center for Health Sciences and Primary Care, 3508 GA Utrecht, The Netherlands; g.w.dalmeijer@umcutrecht.nl

**Keywords:** adolescents, child health, chronic disease, cohort study, COVID-19, mental wellbeing

## Abstract

The aim of this study was to assess the impact of the COVID-19 pandemic on the mental wellbeing of children 8–18 years old with chronic conditions, by comparing pandemic data with pre-pandemic data and with healthy peers. Data were obtained from two ongoing longitudinal cohorts: the PROactive cohort study following children with a chronic condition, and the WHISTLER population cohort. Mental wellbeing was assessed by three indicators: life satisfaction, internalising symptoms, and psychosomatic health. The stringency of the COVID-19-related lockdown was considered a moderating factor. Data on chronic patients were recorded before (n = 934, 65% girls) and during (n = 503, 61% girls) the pandemic, and compared to healthy peers during the pandemic (n = 166, 61% girls). Children with a chronic condition reported lower life satisfaction, but no clinically relevant changes in internalising symptoms or psychosomatic health, during the pandemic compared to before. In comparison to healthy peers, children with a chronic condition experienced decreased life satisfaction and psychosomatic health, but internalising symptoms did not differ between groups during the COVID-19 pandemic. The lockdown stringency was negatively associated with all indicators of mental wellbeing—worse life satisfaction, more internalising symptoms, and more psychosomatic symptoms.

## 1. Introduction

At the end of 2019, a local outbreak of COVID-19 in Wuhan occurred, and rapidly progressed into a global pandemic [1,2]. Governments imposed strict measures to control the spread of the virus, which also impacted the daily routines of children and adolescents (hereafter referred to as children) [3]. The closing of schools and the reduction in social contact with peers are of particular concern from a psychosocial viewpoint [4,5]. Thus, the rapid spread of coronavirus and subsequent social restrictions have led to increased mental health problems [6,7]. The government restrictions differ from country to country. The Oxford Coronavirus Government Response Tracker (OxCGRT) calculates the stringency index, which indicates the strictness of COVID-19 restrictions by day and country [8,9].

COVID-19 literature regarding mental wellbeing has mostly focused on healthy children. The pandemic has often resulted in decreased life satisfaction [10], increased internalising symptoms (including anxiety and depression), and more mental health problems and psychosomatic complaints [11,12,13,14]. However, there is a paucity of knowledge about the effects of the pandemic on children with chronic conditions, who are a population at risk of decreased mental wellbeing [15,16]. In general, children with a chronic condition rate their psychosocial functioning, developmental milestones, and mental wellbeing lower than their healthy peers [15,17]. Pre-existing vulnerabilities—such as socioeconomic disadvantage, elevated levels of internalising and externalising problems, a higher amount of stressful events, or disabilities—are more common in this group, and may increase the risk of poor mental health outcomes during the COVID-19 pandemic [18,19,20,21]. At-risk individuals may experience new onset of mental health problems, while those with pre-existing mental health problems may experience symptomatic exacerbation—especially if access to mental health services is impeded due to COVID-19 regulations [21,22]. To date, empirical studies of the mental wellbeing burden of the pandemic are scarce in this vulnerable population—particularly longitudinal studies [23]. A better understanding is of clinical relevance, since this enables health professionals to incorporate pandemic-related effects into their care for their patients. 

Before and during the pandemic, the Dutch Patient-Reported Outcomes active cohort study (PROactive) [24] and Wheezing Illnesses Study Leidsche Rijn population cohort study (WHISTLER) [25] collected data in children with chronic conditions and healthy peers. Importantly, measurements from both cohorts were harmonized, and participants were recruited from the same geographical area. This provides the unbiased and unique opportunity to study the impact of the pandemic on the wellbeing of children with a chronic condition compared to healthy children [24,26]. Therefore, the aims of this exploratory study were as follows: to compare the mental wellbeing of children with a chronic condition before and during the COVID-19 pandemic (*aim 1*), to compare the mental wellbeing of children with a chronic condition and healthy peers during the pandemic (*aim 2*), and to explore the associations between government restrictions—as measured by the OxCGRT stringency index—and the mental wellbeing of children with a chronic condition and healthy children (*aim 3*). 

## 2. Methods

### 2.1. Research Population and Study Design

We compared mental wellbeing in children with a chronic condition before and during the pandemic (*aim 1*). For this comparison, children aged 8–18 years were selected from the ongoing PROactive cohort study [24,26]. Since 2016, this cohort has been collecting data on psychosocial wellbeing in children with various conditions—including cystic fibrosis, (auto)immune diseases, congenital heart diseases, kidney disease, and persistent physical complaints—visiting the outpatient clinic of the Wilhelmina Children’s Hospital Utrecht, The Netherlands—hereafter referred to as “children with a chronic condition”. Questionnaires completed before March 2020 were classified as “pre-COVID-19”, as the first Dutch case was reported 27 February [27]. All data collected between July 2020 and July 2021 were classified as “during COVID-19”. 

Next, we compared children with a chronic condition with healthy peers, focused on effects during the pandemic (*aim 2*). We used data from the PROactive cohort [24] and the WHISTLER cohort [25]. The WHISTLER cohort’s population included over 3000 newborns residing in the region of Utrecht, The Netherlands, born between 2002 and 2013 [25]. In March 2019, WHISTLER participants were routinely invited to the 12–16-year-old assessments, with a focus on their health and mental wellbeing during adolescence. Due to the onset of the pandemic, they had to pause this follow-up, but the 224 assessments already taken were considered the baseline for a five-wave prospective longitudinal study of changes in mental wellbeing during the Dutch pandemic [10]. For this study, a random sample was drawn from the subsequent 3 waves of data collection during the first year of the Dutch pandemic between 18 July 2020 and 9 March 2021. 

Figure 1 visualises the timing of data collection, along with Dutch COVID-19 restrictions over the course of 15 months. For our 3rd aim, we investigated the effect of the lockdown stringency index on the different indicators of mental wellbeing in children with a chronic condition and their healthy peers [8,9].

Both cohorts were approved by the ethical committee of the University Medical Center Utrecht, The Netherlands. 

### 2.2. Measurements 

Mental wellbeing was assessed using three indicators [10]: life satisfaction, internalising symptoms, and psychosomatic health. Questionnaires were aligned in both the PROactive and WHISTLER studies. Table 1 provides detailed information regarding the measurements.

*Life satisfaction* was measured using the Cantril ladder [30,31], which includes one question: “Looking at the past 3 months, how do you feel about your life?”. Possible answers range from 0 to 10 (10 = best possible life). 

*Internalising symptoms* were assessed using the Revised Child Anxiety and Depression Scale (RCADS) [32,33], which is based on anxiety disorders and depression from the DSM-IV [33]; it is a 47-item questionnaire with anxiety subscales such as social phobia, generalised anxiety disorder, depressive disorder, etc. The sum of all of the subscales (total score) is a global indication of internalising symptoms, with higher scores indicating more severe symptoms. As in our study the correlation between the subdomains “anxiety” and “depressive disorder” was r > 0.7, we choose to analyse these subdomains together as internalising symptoms. Based on age and sex, raw scores were converted to normative T-scores [34]. A score <65 is considered normal, 65–70 is borderline, and >70 is critical. 

*Psychosomatic health* was assessed using the Dutch Health Behaviour in School-Aged Children Symptom Checklist (HBSC-SCL) 2017 [35,36], consisting of 10 questions evaluating the severity of symptoms, such as having a headache, being nervous, etc. These symptoms are often related to psychosocial factors, such as stress [10,37]. A high mean score reflects better psychosomatic health [15]. This instrument (Dutch 2017 version) has good psychometric properties, and has been validated as an unbiased measurement of subjective health complaints (Cronbach’s alpha > 0.70) [15].

*Stringency index* was assessed with the OxCGRT [8,9], providing the stringency of COVID-19 restrictions per day and country. The index is based on 23 indicators, such as school closures and travel restrictions, resulting in a score of 0 to 100 (100 = strictest) [9]. We linked this stringency scores to the dates the patients’ completed the questionnaires.

### 2.3. Statistical Analyses

We compared the mental wellbeing of children with a chronic condition before and during the pandemic (*aim 1*), as well as mental wellbeing between children with a chronic condition and healthy peers during the pandemic (*aim 2*), using analysis of variance (ANOVA). Here, we considered two independent variables (main effects) and their interaction: time point (*aim 1*) or cohort (*aim 2*), and gender (girls or boys), as well as the interactions time point*gender (*aim 1*) and group*gender (*aim 2*). The interaction provides information on the extent to which potential gender differences are similar between time points (*aim 1*) and groups (*aim 2*). In case of significant interactions, a stratified ANOVA was performed. In case of significant difference in time points (*aim 1*), the mean differences from aim 1 before and during the pandemic of the WHISTLER cohort [10] were compared (data not shown). We explored the association between local government restrictions (stringency index) and the mental wellbeing of children with a chronic condition and healthy children using a hierarchical linear regression (*aim 3*). In the hierarchical linear regression, we entered the stringency index, group (children with a chronic condition or healthy peers), and the interaction stringency index*group as independent variables in steps 1, 2, and 3, respectively. Separate models were run for each of the dependent variables (i.e., life satisfaction, internalising symptoms, and psychosomatic health). An observed *p*-value of <0.05 was considered statistically significant.

## 3. Results

### 3.1. 1st Aim; Mental Wellbeing in Children with a Chronic Condition before versus during the Pandemic

Table 2 shows the characteristics of the two PROactive cohort samples of children with a chronic condition before (n = 944) and during the pandemic (n = 545). These are two different samples of children with a chronic condition. Life satisfaction was significantly lower during the pandemic compared to before the pandemic (F(1, 1468) = 30.27; *p* < 0.001). Girls had a significantly lower life satisfaction score compared to boys (F(1, 1469) = 42.70; *p* < 0.001). The interaction time point*gender was not significant (F(1, 1469) = 2.25; *p* = 0.13), indicating that COVID-19 had no difference in impact on girls than on boys. Figure 2 visualizes the findings. 

We found non-significant findings for the two remaining dependent variables. First, internalising symptoms were similar during the pandemic compared to before the pandemic (F(1, 1151) = 0.00; *p* 0.96). Girls experienced internalising symptoms significantly more often compared to boys (F(1, 1151) = 43.24; *p* ≤ 0.001), although in neither group was the change in mean score clinically relevant (mean < 60). The interaction time point*gender was not significant (F(1, 341) = 2.92 *p* 0.09). Second, psychosomatic health did not significantly differ during the pandemic compared to before the pandemic (F(1, 1151) = 0.00; *p* 0.40). Girls experienced more psychosomatic symptoms (F(1, 1151) = 4.58; *p* ≤ 0.01). The interaction time point*gender was not significant (F(1, 1151) = 1.223; *p* 0.27), indicating that similar differences between genders were observed for both time points; thus, stratified analyses were not necessary.

### 3.2. 2nd Aim: Mental Wellbeing in Children with a Chronic Condition versus Healthy Peers during the Pandemic

For the second aim, children with a chronic condition (n = 311) were compared to healthy peers (n = 166) during the pandemic. Table 2 provides the children’s characteristics. Life satisfaction was significantly lower in children with a chronic condition compared to healthy peers (F(1, 473) = 13.92; *p* < 0.001). Girls reported lower life satisfaction than boys (F(1, 473)= 42.70; *p* < 0.001). The interaction group*gender was not significant (F(1, 473) = 0.05; *p* = 0.83), indicating that the reported difference is attributable to their chronic condition. Figure 3 visualises the findings.

Additional analysis with pre-pandemic WHISTLER data showed a difference in both cohorts (PROactive and WHISTLER) of 0.5 points in life satisfaction before and during the pandemic (data not shown), indicating that children with a chronic condition experienced a similar effect compared to healthy peers. 

Psychosomatic health complaints were reported significantly more often in children with a chronic condition compared to healthy peers (F(1, 345) = 91.77; *p* ≤ 0.001). Girls experienced more psychosomatic symptoms (F(1, 345) = 48.48; *p* ≤ 0.001) than boys. The interaction group*gender was not significant (F(1, 345) = 1.76; *p* ≤ 0.19). 

The internalising symptoms score was not significantly different in children with a chronic condition compared to healthy peers (F(1, 359) = 0.026; *p* 0.87). Girls had more internalising symptoms (F(1, 359) = 26.40; *p* 0.00) than boys. The interaction time group*gender was not significant (F(1, 359) = 2.50; *p* 0.12). 

### 3.3. 3rd Aim: Associations between Government Restrictions and the Mental Wellbeing of Children with a Chronic Condition and Healthy Children

Here, a hierarchical linear regression analysis with separate models was conducted for each of the dependent variables (see Table 2). In step 1, higher scores on the stringency index were associated with worse life satisfaction (t = −3.67, *p*  < 0.001), internalising symptoms (t = 3303, *p* ≤ 0.001), and psychosomatic health (t = −2.48, *p* ≤ 0.01). Stringency index explained 2–3% of the variability in mental wellbeing; adjusted R^2^ values were 0.03, (F(1, 475) = 13.47; *p*  < 0.001) for life satisfaction, 0.03, (F(1, 360) = 6.42; *p*  < 0.002) for internalising symptoms, and 0.02, (F(1, 347) = 6.16; *p* 0.01) for psychosomatic health. In step 2, disease state (i.e., chronic condition vs. healthy peers) additionally explained 3–6% of the remaining variability in general wellbeing; adjusted R^2^ values were 0.06, (F(1, 474) = 20.44; *p*  < 0.001) for life satisfaction, 0.04, (F(1, 360) = 4.59; *p* 0.04) for internalising symptoms, and 0.27, (F(1, 346) = 121.39; *p* 0.00) for psychosomatic health. In step 3, stringency index, group (disease state), and the interaction stringency*group explained 28% of the variability in the dependent variable “psychosomatic health” (adjusted R^2^ = 0.28; F(1, 345) = 7.78; *p* 0.01). The interaction stringency index*group was not significant for life satisfaction nor for internalising symptoms (*p* 0.68 and 0.34). Table 2 displays the findings.

## 4. Discussion

This study of children’s mental wellbeing compared pandemic data with pre-pandemic data between children with a chronic condition and healthy peers. The present study provides four key findings: First, the pandemic had a negative impact on the life satisfaction of children with a chronic condition, but our data showed no clinically relevant changes in internalising symptoms or psychosomatic health during the pandemic compared to before. Second, compared to healthy peers, children with a chronic condition experienced poorer life satisfaction and psychosomatic health during the pandemic, but internalising symptoms did not differ between groups. Third, compared to boys, girls robustly reported worse mental wellbeing, and this difference was apparent regardless of the pandemic or their disease state. Fourth, stricter governmental restrictions [9] were significantly associated with poorer life satisfaction, more internalising symptoms, and worse psychosomatic health in both children with a chronic condition and healthy peers, with the stringency index explaining up to 28% of variance in psychosomatic symptoms. 

Adolescents with a chronic condition might be particularly susceptible to the effects of the pandemic on mental wellbeing. Our study found that, compared with healthy peers, children with a chronic condition experienced both decreased life satisfaction and decreased psychosomatic health during the pandemic; however, this difference was probably pre-existing. We performed an additional analysis with pre-pandemic WHISTLER data that showed a mean difference in life satisfaction in both cohorts of 0.5 points before and during the pandemic. This suggests that children with a chronic condition did not experience more distress than healthy peers due to the pandemic, but that the difference was there before, and remained without increasing.

Previous studies of mental wellbeing in children during and before the pandemic can be used to benchmark our findings, keeping in mind that these studies likely did not evaluate mental wellbeing throughout the first year of the pandemic, and that little literature is available on the impact of the pandemic on wellbeing in children with a chronic condition.

Zijlmans et al. [38] compared a clinical paediatric sample (aged 8–18 years, n = 90, including juvenile idiopathic arthritis, endocrinological diseases, and cystic fibrosis) with the general population (n = 844), and reported significantly better scores for anxiety, depressive symptoms, and anger in the clinical sample; however, they collected data in a relatively small group and small timeframe at the beginning of the Dutch pandemic (April–May 2020), making their data difficult to compare with our data. Nevertheless, children with pre-existing mental health problems had lower mental wellbeing than children with somatic disease or healthy peers. The findings suggest that it is possible that some children growing up in more challenging circumstances—such as those with a chronic illness—are more resilient than healthy children. Therefore, follow-up research within our research field should aim to identify what risk and resilience factors might influence changes in mental wellbeing during the pandemic—especially in subgroups at higher risk of mental health problems. As a result, we hope that when another pandemic occurs, we will have a better understanding of which children need additional observations and support. A recent study in children with genetic generalised epilepsy showed emotional and psychological resilience during the COVID-19 pandemic; we wonder whether this is also the case transdiagnostically (across disease groups) and in comparison to healthy peers [39]. A recent review of 116 articles that evaluated the impact of the pandemic on the mental health of children—including children with a chronic condition—concluded that children with neurodiversity and/or chronic physical conditions were more likely to have negative mental health outcomes such as fear, anxiety, and depression compared to healthy peers [40]. This is not consistent with our findings, as we found no changes in internalising symptoms in children with a chronic condition. This may be due to differences in outcome measures, as well as the relatively small numbers of children with a chronic condition in this review. Notably, less than 15% of the available studies in this review used validated instruments, which the authors rightly state leads to challenges in interpreting the clinical relevance of mental health impacts and differentiation between adaptive symptoms and mental illness. Neither study considered the relationship between mental health outcomes and the degree of governmental restrictions. 

We showed that a stricter OxCGRT stringency index [9] was associated with worse life satisfaction, more internalising symptoms, and worse psychosomatic health in both children with a chronic condition and healthy peers. These data suggest that distress is associated with the degree of governmental restrictions. Our results are consistent with a recent systematic review [41] that reported on the association of school closures during the broader social lockdown of the first waves of the pandemic with mental health, health behaviours, and wellbeing in children aged 0–19 years. The authors found that school closures and social lockdown during the first wave of the pandemic were associated with adverse mental health symptoms (such as anxiety and distress) and health behaviours (such as reduced physical activity and more screen time) [41]; they could not distinguish between the effects of school closures and broader social lockdown measures.

Our and their findings support the idea that the potential epidemiological benefits of closing schools during broader social lockdown measures for infectious disease control must be weighed against the potential adverse effects on mental wellbeing and health behaviours in children. These findings are important for informing government and society about the adverse impacts of the pandemic on children’s mental wellbeing with regard to closure measure choices, and also advocate the use of the stringency index in this type of study.

Some strengths and limitations deserve consideration. Our findings are novel and exploratory in nature; independent replication by other research teams would greatly strengthen the conclusions. Our data do not allow us to identify links between different lockdown measures—for example, between school closures/social distancing and mental wellbeing outcomes—nor was it possible to identify which children are most at risk of adverse mental health outcomes. Therefore, it is of interest for future research to consider the extent of governmental restrictions in studies that include mental wellbeing as an outcome measure for the impact of the pandemic, as well as to identify risk and resilience factors that may influence the impact of the pandemic on mental wellbeing in children. The inclusion of two cohorts from the same geographical area and with harmonised measurements is a strength of the present study. Additional strengths include the relatively large sample size and the inclusion of several indicators of mental wellbeing. An interesting future directive would be to substantiate the recorded subjective information with objective biomarkers (e.g., cortisol as a marker of stress), as this may provide a more comprehensive understanding of our findings. 

## 5. Conclusions 

To conclude, children with a chronic condition reported lower life satisfaction during the pandemic than before the pandemic. Compared to healthy peers, both life satisfaction and psychosomatic health were worse in children with a chronic condition. COVID-19 governmental restrictions were associated with all indicators of mental wellbeing, and explained up to 28% of the observed variation in both children with a chronic condition and healthy peers. Further research should focus on determining the clinical relevance of these findings, and explore strategies to identify those children most at risk of serious deterioration in mental wellbeing.

## Figures and Tables

**Figure 1 ijerph-19-02953-f001:**
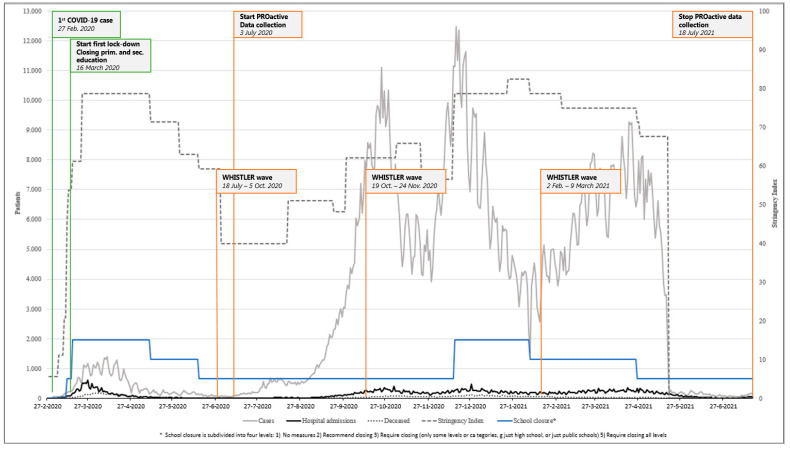
Prevalence of confirmed COVID-19 cases [28], hospital admissions [28], deaths [28], school closures [28], and stringency index scores [29], along with specific time points of the present study (data originating from the Dutch National Institute for Public Health and the Environment (RIVM)).

**Figure 2 ijerph-19-02953-f002:**
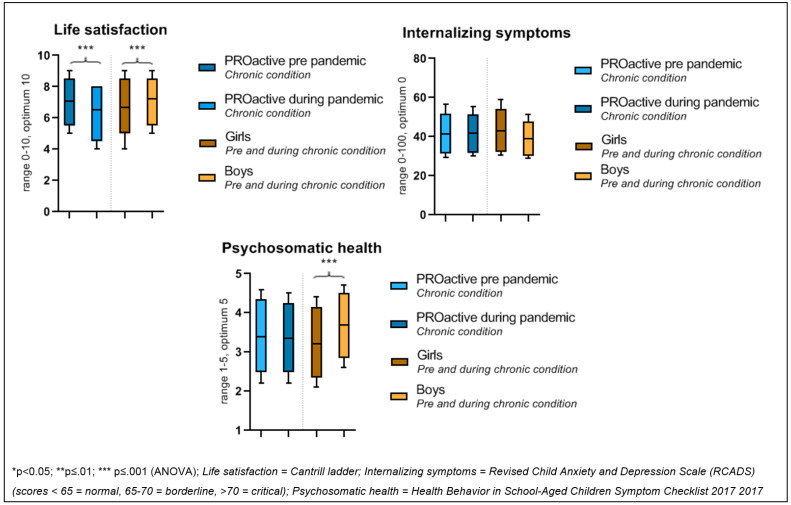
Group difference indicators of mental wellbeing in children with a chronic condition before and during the COVID-19 pandemic (*aim 1*).

**Figure 3 ijerph-19-02953-f003:**
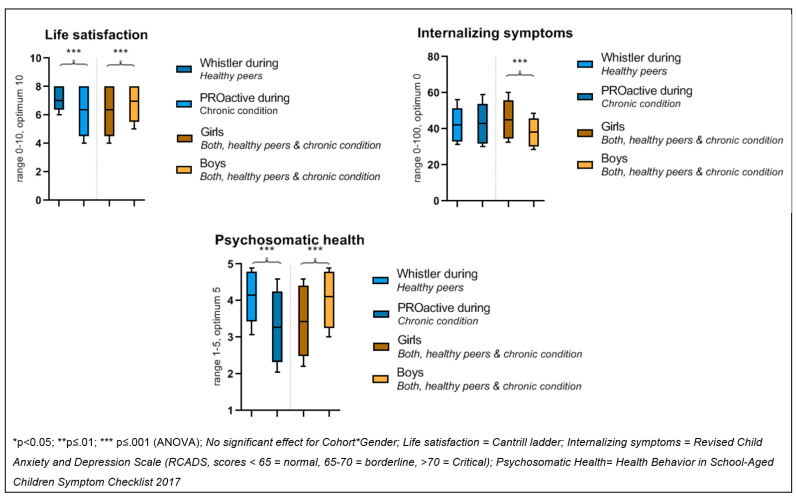
Group difference indicators of mental wellbeing in children with a chronic condition versus healthy peers during the pandemic (*aim 2*).

**Table 1 ijerph-19-02953-t001:** Characteristics of the participants.

	1st Aim (Chronic Condition before vs. during)		2nd and 3rd Aims (Chronic Condition vs. Healthy Peers during the Pandemic)	
Characteristics	PROactive (Chronic Condition)		PROactive—(Chronic Condition) ^a^	Whistler—(Healthy Peers)
	*Pre-Pandemic* *8–18 yr* *N = 944*	*During Pandemic* *8–18 yr* *N = 545*	*During Pandemic* *12–18 yr* *N = 311*	*During Pandemic* *12–18 yr* *N = 166*
Age in years, *mean ± SD*	14.2 (2.8)	14.3 (2.9)	15.7 (1.8)	16.0 (1.3)
Girl, n (%)	601 (64.2)	332 (60.9)	200 (64.3)	94 (56.6)
Disease group				
- Persistent physical complaints	481 (51.0)	269 (49.4)	67 (21.5)	n.a.
- Paediatric (auto)immune diseases	305 (32.3)	106 (19.4)	163 (52.4)	n.a.
- Paediatric cystic fibrosis	72 (7.6)	26 (4.8)	17 (5.5)	n.a.
- Paediatric cardiology	63 (6.7)	111 (20.4)	46 (14.8)	n.a.
- Paediatric nephrology	22 (2.3)	33 (6.1)	18 (5.8)	n.a.
Education level of the child ^b^, n (%)				
- Primary school	219 (24.0)	105 (25.6)	14 (5.8)	3 (1.8)
- Low	271 (29.7)	117 (28.5)	94 (39.0)	40 (24.1)
- Intermediate	208 (22.8)	86 (21)	69 (28.6)	37 (22.3)
- High	177 (19.4)	79 (19.3)	64 (26.6)	81 (48.8)
- Other (special education or working)	37 (4.1)	23 (5.6)	0 (0)	5 (3.0)

^a^ For comparison with healthy peers (*aim 2 and 3*), the 12–18-year-old children from the PROactive cohort were selected. ^b^ Low: pre-vocational secondary education; intermediate: higher general secondary education or intermediate vocational education; high: pre-university education, higher vocational education, and university education. SD: standard deviation.

**Table 2 ijerph-19-02953-t002:** Regression indicators of mental wellbeing in children with a chronic condition versus healthy peers during the pandemic (*aim 2 and 3*).

		Life Satisfaction					Internalising Symptoms					Psychosomatic Health				
		B	95% Confidence Interval for B		β	Adj. R^2^	B	95% Confidence Interval for B		β	Adj. R^2^	B	95% Confidence Interval for B		β	Adj. R^2^
			Lower	Upper				Lower	Upper				Lower	Upper		
Step 1						0.026 ***					0.029 *					0.015 *
	Stringency	−0.015	−0.023	−0.007	−0.166 ***		0.107	0.043	0.171	0.183 ***		−0.007	−0.012	−0.001	−0.132 **	
Step 2						0.064 ***					0.034 **					0.268 ***
	Stringency	−0.017	−0.025	−0.009	−0.185 ***		0.114	0.050	0.178	0.183 ***		−0.009	−0.014	−0.005	−0.182 ***	
	Group	−0.698	−1.002	−0.395	−0.201 ***		1.683	−0.718	4.08	0.072 ^ns^		−0.912	−1.075	−0.749	−0.508 ***	
Step 3						0.062 ^ns^					0.037 **					0.282 **
	Stringency	−0.010	−0.042	0.021	−0.114 ^ns^		0.000	−0.241	0.0.241	0.001 ^ns^		0.12	−0.004	−0.027	0.238 ^ns^	
	Group	−0.693	−0.998	0.389	−0.200 ***		1.529	−0.892	3.951	0.066 ^ns^		−0.900	−1.061	−0.738	−0.501 ***	
	Interaction	−0.004	−0.021	0.041	−0.073 ^ns^		0.068	−0.071	0.206	0.188 ^ns^		−0.013	−0.022	−0.004	−0.438 **	

B: unstandardized regression coefficient; β: standardized regression; Adj. R^2^: adjusted R^2^ with significance levels of F-change; ^ns^: *p* not significant; *: *p* < 0.05; **: *p* ≤ 0.01; ***: *p* ≤ 0.001.

## Data Availability

The data presented in this study are available on request from the corresponding author. See our data request procedure on DataverseNL https://doi.org/10.34894/FXUGHW (accessed on 31 August 2020).

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
