# Peer review of "The Impact of the COVID-19 Outbreak on Mental Wellbeing in Children with a Chronic Condition Compared to Healthy Peers"

_ijerph, 2022, doi:10.3390/ijerph19052953_

Round 1

Reviewer 1 Report

I think that the combination of data do give an interesting way to compare and see pre and during covid. And that makes the paper interesting. 
For me I see that perhaps the chronic condition or disability group is not as usually focused on. For me, being from that field, some data are no news, and I would have liked to see some more insight into research actually showing that this will have lower mental health and well being than typically developing children and youth. Disability is a common riskfactor, So yes, they did have lower to start with, something you do say in the introduction and also in discussion. The argument is, only, that children with a chronic condition already have a risk for decreased mental wellbeing, not why they would be more affected or at the same level as everyone else. Why would this group differ. I could think of several things so it should be easy to quickly argue for this.

Chronically ill (are you always ill if you have a condition?), also health peers, are you healthy if you do not have a condition? The choosing of words are a bit normative. Perhaps typical and chronic conditions could be used? Also terminology as disease state, morally it really sounds bad. Chronic condition or disability would be preferred.

Tables should speak for themselves, when you are talking about educaional level, you mean parental education? Are you using this information for something?

Psychosomatic complaints,  the scale in HBSC have 5 response categories, (1-5). And a higher value being lower levels of symptoms. So that fit with how you report it. but not 0-5? Also looking at your supplement, "feeling unhappy and having a bad mood, are not worded the same way as HBSC, and the questions being tired and being exhausted are not a part of HBSC, according to their protocol. Therefore can you argue that it is validated as HBSC scale? 

In the figure on page 6. You report life satisfaction both groups do you mean both those with and without disabilities? Just as those divided by chronically ill (disabilities) would be both gender? Or are you referring to both before and during pandemic? This is not clear from figure. reading the text I think I get it but perhaps the wording in figures could be better? 

Reviewer 2 Report

Please add an introduction to the trials whose data was used for this research study.

Reviewer 3 Report

Thank you very much for allowing me to review the work. First of all, I would like to stress that the authors have done a good job.

I would like to make a number of recommendations to improve the quality of the work:
The Introduction has relevant information, but I consider that it is necessary to expand it with more previous literature that allows us to identify the importance of the work and the objectives.

The Methodology is well presented and it seems that the treatment of the data, as well as the results obtained, are coherent with the approach of the study.

In the Discussion, it would be advisable to compare the results with more previous studies, since although this is a new topic, there is already a lot of literature on the field of study.

Finally, I recommend including practical applications and future lines of research, which will add value to the work.

Reviewer 4 Report

The article brings insight into the mental wellbeing of covid affected childre. please improve the discussion by analysing the impact of antivirals used in covid infection on mental health. please check: https://www.sciencedirect.com/science/article/pii/S0753332222000889 â

also describe the impact of covid in patients with diabetes mellitus please check: Bondar A, Popa AR, Papanas N, Popoviciu M, Vesa CM, Sabau M, Daina C, Stoica RA, Katsiki N, Stoian AP, Stoian AP, et al: Diabetic neuropathy: A narrative review of risk factors, classification, screening and current pathogenic treatment options (Review). Exp Ther Med 22: 690, 2021.

Minor revision
